# Consumption of Tap Water and Sociodemographic-Associated Characteristics: A Nationwide Cross-Sectional Study

**DOI:** 10.3390/nu16070944

**Published:** 2024-03-25

**Authors:** Jacopo Dolcini, Manuela Chiavarini, Elisa Ponzio, Giorgio Firmani, Marcello Mario D’Errico, Pamela Barbadoro

**Affiliations:** 1Department of Biomedical Sciences and Public Health, Section of Hygiene, Preventive Medicine and Public Health, Polytechnic University of the Marche Region, 60131 Ancona, Italy; m.chiavarini@staff.univpm.it (M.C.); e.ponzio@staff.univpm.it (E.P.); g.firmani@pm.univpm.it (G.F.); m.m.derrico@staff.univpm.it (M.M.D.); 2Centre of Obesity, Marche Polytechnic University, Via Tronto 10a, 60126 Ancona, Italy

**Keywords:** safe water, public health, sociodemographic characteristics, environmental health, health outcomes

## Abstract

Safe water is a global public health concern amid increasing scarcity and pollution. Bottled water production and consumption contribute to these problems. This study examines tap water consumption in Italy, assessing associated sociodemographic factors and related health outcomes such as obesity and self-perceived health status. Data from the Italian National Statistics Institute’s “Aspects of daily life” survey (N = 45,597) were analyzed. Covariates included education, age, gender, economic status, region, concerns about waste and climate change, consumption of carbonated drinks excluding water, alcohol consumption, consumption of vegetables, consumption of snacks, body mass index, and self-perceived health status. Bivariate analyses and mixed-effect logistic regression models explored the associations. People who drink tap water made up 19,674, with a higher prevalence in people aged 45 to 59 old, people with a graduate/post-graduate degree diploma, with optimal economic resources, people concerned about waste production and climate change, and those coming from the north-east regions of Italy. Underweight people showed a higher prevalence of TW consumption as well as those who less than occasionally drank carbonated drinks, drank alcohol, consumed vegetables more than once a day and snacks less than once a week, dairy products more than once a day, sweet less than once a week, cured meat less than once a week, and chicken meat less than once a week, those with no consumption of sheep meat, consumption of beef meat less than once a week and consumption of pork meat less than once a week, and those with a satisfactory level of perceived health status. Regressions showed that all other age classes are less likely to drink tap water than people younger than 20 years old. The category with “inadequate” economic resources is more likely to consume tap water. Low educational classes show a low likelihood of consuming tap water as well as islands. A concern about waste production and climate change is associated with an increased likelihood of consuming tap water. Tap water consumption was negatively associated with obesity but not with a satisfactory self-perceived health status. Insights from this study can inform public health strategies.

## 1. Introduction

Drinking safe water is a critical public health element worldwide. Improved water supply and sanitation, and better management of water resources, can boost countries’ economic growth and can contribute greatly to poverty reduction [1]. Anyway, in the next future, the availability of safe water could be a worldwide issue [2,3]. Environmental challenges such as climate change and extreme events, water scarcity, and pollution are increasingly becoming a global concern [4,5,6]. Many studies have shown how the production and consumption of bottled water (BW) are contributors to these problems [3,4]. Others have shown how tap water (TW) would be preferable due to the high environmental impact of bottle manufacturing [7]. Moreover, increasing evidence has shown a higher level of exposure to microplastics from BW versus TW [8], increasing concerns about possible human health effects from its consumption. Nowadays, in many countries, there is the possibility to have access to cheap and safe potable TW, but BW consumption has been increasing globally [9]. Recently, there has been a growing interest in investigating possible reasons for preferring TW or BW. Some of them have focused attention on health and safety concerns about TW [8,9], while others on organoleptic properties [10,11,12]. The European region has diverse TW consumption patterns between and within countries, with the highest share of TW intake observed in northern Europe [13]. Moreover, different sociodemographic characteristics associated with TW or BW consumption have been investigated, delineating a complex scenario [14]; in particular, gender differences have been highlighted, with women drinking more BW [13,14,15] since women seem to perceive TW riskier but at the same time, they have more general environmental awareness [16,17]. Moreover, ethnicity and culture, as well as self-perceived health status and diet choices are among the key drivers of TW and city dwellers tend to drink less TW, with the lowest consumption observed at restaurants and the highest in people’s homes [13,18]. Other studies have found that the consumption of TW was higher at higher incomes [19] while age showed conflicting results, with older people drinking more TW than others in some areas while the opposite is true in others [20,21]. Regarding education, some studies showed that individuals with higher education are less likely to drink bottled water as their primary source of water [20] and general lower educational attainment showed a higher prevalence of TW avoidance among adults and children [14]. Additionally, research shows that free access to safe and acceptable drinking water may be beneficial in reducing sugary drinks [22] and also it has an important role in weight reduction, body fat reduction, and appetite suppression and it may be useful in the prevention of overweight and obesity [23,24], but the equation with health outcomes may be complicated by complex scenarios where in some regions TW consumption may be influenced by increased exposure to actual and perceived risk of endocrine-disrupting compounds [25].

The aim of this study is to evaluate the consumption of TW and its association with selected sociodemographic characteristics and dietary habits at a nationwide level in Italy.

Moreover, we evaluated the possible association between TW consumption and health-related outcomes, particularly focusing on self-perceived health status and an obese Body Mass Index.

## 2. Materials and Methods

(i)Design, Data Source, and Participants

Information on tap and BW consumption was obtained from the multi-purpose survey on families’ “Aspects of daily life”, carried out by the Italian National Statistics Institute (ISTAT) [26]. The survey is carried out on a yearly basis in a representative sample of the Italian population and is part of the integrated system of multipurpose surveys on families aimed at detecting a plurality of behavioral dimensions and segments of daily life. The questionnaire is standardized, and the survey is carried out by a sequential CAWI/PAPI mixed-mode technique [26] and it has been used in several studies [27,28]. We analyzed data coming from the 2021 edition of the survey, including 45,597 subjects and 20,000 families. In this specific case, the populations for the current surveys were identified by the Italian National Statistics Institute within the set of municipalities, which was divided into two subsets: municipalities with larger demographic size constituted a separate stratum and were defined as Self-Representative (SR); the remaining municipalities were defined as Non Self-Representative (NSR) and were divided, based on demographic size, into strata of equal breadth. From these strata, the sample municipalities (two for each stratum) were selected with probabilities proportional to their size. For each of the municipalities involved in the survey (SR and NSR), a cluster sampling was carried out: the clusters—the families—were randomly selected from the registry list, and all the members belonging to the actual family were surveyed. The minimum number of sample families for each municipality was set to 24. The families were selected for each sample municipality from the theoretical sample selected for the Master Sample for each family included in the sample, and the characteristics under investigation of all actual members belonging to the same family were recorded. The size of the theoretical sample in terms of families, set at the national level primarily based on cost and operational criteria, was approximately 24,000 families. The number of involved sample municipalities should not exceed 900, to allow for effective control and supervision. The allocation of the sample of families and municipalities, among the various regions, was therefore calculated by adopting a compromise criterion to ensure both the reliability of estimates at the national level and that of estimates within each of the territorial domains [26].

(ii)Variables

The following variables were included in the analyses: educational level (graduate/postgraduate degree, high school diploma, middle school diploma, primary school diploma/none), age (<20 years old, 20–44 years old, 45–59 years old, 60–74 years old, ≥75 years old), gender (males/females), economical resources in last 12 months (1 = inadequate, 2 = scarce, 3 = adequate, and 4 = optimal), geographical area of residency (north-west, north-east, center, south, the islands), concern about waste production (yes/no), concern about climate change (yes/no), consumption of carbonated drinks excluding water (1 = more than 1 L/day, 2 = between ½/1 L day, 3 = less than ½ L, 4 = less than occasionally, 5 = occasionally, 6 = no use), alcohol consumption (1 = yes, 0 = no), consumption of vegetables (1 = more than once a day, 2 = once a day, 3 = sometimes a week, 4 = less than once a week, 5 = never), consumption of snacks (1 = more than once a day, 2 = once a day, 3 = sometimes a week, 4 = less than once a week, 5 = never), body mass index (BMI; 1 = underweight (<18.5 kg/m^2^), 2 = normal (18.5–24.99 kg/m^2^), 3 overweight (25.00–29.99 kg/m^2^, 4 = obese (>30.00 kg/m^2^), self-perceived health status (1 = satisfactory, 0 = not satisfactory), and consumption of dairy products, sweet, cured meat, chicken meat, sheep meat, beef meat, and pork meat (1 = more than once a day, 2 = once a day, 3 = sometimes a week, 4 = less than once a week, 5 = never). 

(iii)Ethical Considerations

Ethical review and approval were waived for this study since we utilized data from the Italian Health Interview Survey, a routine initiative conducted by the Italian National Institute of Statistics according to national guidelines. As a national institute operating under legal mandates, ethical approval is not mandated for nationwide surveys. The data underwent anonymization and aggregation by ISTAT, ensuring privacy, and were subsequently made publicly accessible for research purposes. Informed consent was obtained from all subjects involved in the study.

(iv)Statistical Analysis

Bivariate analyses were performed to study the association of TW consumption with relevant variables using chi-square tests. Logistic regression models were built to adjust for confounders and to evaluate the factors independently associated with TW consumption (1, if TW consumption is present; 0 if not). The Hosmer–Lemeshow test was used to evaluate the goodness of fit of the model. Regarding health outcomes, we considered self-perceived health status satisfaction level (dichotomized in two levels: 1 = satisfactory, 0 = not satisfactory) and BMI (dichotomized in two levels: 1 = obese, 0 = not obese). The level of significance was set to 0.05. Analyses were performed with STATA, version 15 (Stata Corp., College Station, TX, USA).

## 3. Results 

(i)Bivariate analysis

Among the selected sample, people who usually drink TW were 19,674, representing 43.15% (95% confidence interval, CI 42.69–43.60). In terms of prevalence, bivariate analysis (Table 1) highlighted a higher prevalence of TW consumption in people aged 45 to 59 years old (N = 4942, 25.12%, *p*-value < 0.05) with no statistically significant differences among males and females. People with a graduate/post-graduate degree diploma showed the highest prevalence of consumption (N = 3111, 48.56%, *p*-value < 0.05) as well as people with optimal economic resources (N = 356, 50%, *p*-value < 0.05). Considering concern for environmental issues, our sample showed a higher prevalence of people consuming TW among those that were concerned about waste production (N = 7963, 44.21%, *p*-value < 0.05) and among those that were concerned about climate change (N = 9586, 44.66%, *p*-value < 0.05). Regarding the Geographical Area of Residency, there was a higher prevalence of TW consumption in the north-east regions of Italy (N = 5966, 64.98%, *p*-value < 0.05). Underweight people showed higher prevalence of TW consumption (N = 509, 46.27% *p*-value < 0.05), as well as those who drank less than occasionally carbonated drinks (N = 1932, *p*-value < 0.05); drank alcohol (N = 12,519, 44.66%, *p*-value < 0.05); consumed vegetables more than once a day (N = 3584, 46.83%, *p*-value < 0.05), snacks less than once a week (N = 7265, 45.10%, *p*-value = 0.05), dairy products more than once day (N = 910, 48.74, *p*-value < 0.05); sweets less than once a week (N = 7372, 43.68%, *p*-value = 0.25), cured meat less than once a week (N = 5363, 44.58%, *p*-value < 0.05), and chicken meat less than once a week (N = 3393, 48.5%, *p*-value < 0.05); never consumed sheep meat (N = 9126 45.78%, *p*-value < 0.05); consumed beef meat less than once a week (N = 6470 47.15%, *p*-value < 0.05) and consumed pork meat less than once a week (N = 7690, 44.99%, *p*-value < 0.05). Regarding perceived health status, the highest prevalence of TW consumption was shown from those with a satisfactory level (N = 14,323, 44.03%, *p*-value < 0.05).

(ii)Logistic regression for TW consumption

Multilevel regression analysis predicting the level of one of the dependent variables, which was the presence of TW consumption (1 = presence of TW consumption), showed that (Table 2) all age classes from 20 to 74 years old were less likely to drink TW than people younger than 20 years old but only two classes reached statistical significance (20–44 OR = 0.75, C.I. 0.65–0.88, 45–59 OR = 0.85, C.I. 0.73–0.99). Our model highlights also how the level of schooling may influence TW consumption. Compared to the highest level of education (graduate/post-graduate), other educational classes showed a low likelihood of consuming TW, with a statically significant level (High School diploma OR = 0.8, C.I. 0.75–0.85, Middle School Diploma OR = 0.71, C.I. 0.66–0.77, Elementary school diploma/none OR = 0.69, C.I. 0.63–0.76). Regarding the Geographical Area of residence, we did observe statically significant differences among different regions, with increased consumption in the north-west (baseline) and north-east (OR = 2.13, C.I. 1.99–2.27) and less consumption in the center (OR = 0.85, C.I. 0.8–0.91), south (OR = 0.67, C.I. 0.63–0.72) and the islands (OR 0.25, C.I. 0.22–0.27). In our model, we also considered some variables associated with attention to environmental issues. A concern about waste production was associated with an increased likelihood of consuming TW (OR = 1.09, C.I. 1.04–1.14). A similar trend of likelihood was shown for concern for climate change but was not statistically significant (OR = 1.05, C.I. 1.00–1.1). Looking at eating habits, TW consumption was associated with alcohol consumption (OR = 1.11, C.I. 1.05–1.17), with the lowest classes of sweet consumption (“less than once a week” OR = 1.22 C.I. 1.03–1.45, “never” 1.27 C.I. 1.06–1.52) and the lowest classes of chicken and beef meat (respectively “less than once a week” OR = 1.45 C.I. 1.16–1.81, “never” OR = 1.33 C.I. 1.04–1.70; “less than once a week” OR = 1.67 C.I. 1.13–2.45, “never” OR = 1.58 C.I. 1.06–2.33). Regarding dairy products and sheep meat consumption, all decreasing classes of consumption showed a low likelihood of TW consumption being statically significant. 

(iii)Logistic regression for perceived health status

We then performed a multilevel regression analysis predicting the level one of the dependent variables, the presence of satisfactory self-perceived health status (1 = satisfactory perceived health status), with perceived health status as the main outcome (Table 3). One possible variable that was independently associated, the consumption of tap water, showed no statistically significant association (OR = 1.05, C.I. 0.99–1.11) with satisfactory perceived health status. This perception showed a decreasing trend of likelihood with age increase, and it reached statistical significance among all different age classes (20–44, OR = 0.65, C.I. 0.49–0.88; 45–59, OR = 0.34, C.I. 0.25–0.45; 60–74, OR = 0.21, C.I. 0.16–0.28; ≥75 OR = 0.12, C.I. 0.08–0.15). The same decreasing trend was found regarding lower levels of education, but only people with a middle school diploma (OR = 0.90, C.I. 0.81–0.99) and with an elementary school diploma/no diploma (OR = 0.66, C.I. 0.59–0.74) reached statistical significance. A low likelihood of a satisfactory perceived health status was found also with a decreasing availability of economic resources in the last 12 months, particularly for people who declared few (OR = 0.51, C.I. 0.39–0.66) and insufficient resources (OR = 0.33, C.I. = 0.25–0.45). Again, self-perceived health status was worse in the central, southern and island regions (OR = 0.88, C.I. 0.8–0.96; OR = 0.8, C.I. 0.73–0.87; OR = 0.73, C.I. 0.66–0.81), as well as in underweight (OR = 1.47, C.I. 1.23–1.75) and overweight people (OR = 1.32, C.I. 1.10–1.58) with respect to normal-weight subjects. Regarding the presence of alcohol consumption, it was found to be positively associated with perceived health status (OR = 1.32, C.I. 1.24–1.41). Considering the consumption of vegetables, the likelihood of association with a satisfactory perceived health status showed opposite trends since the lower the consumption of vegetables, the lowest odds of association, and this was significant for “sometimes a week” (OR = 0.88, C.I. 0.81–0.96), “less than once a week” (OR = 0.74, C.I. 0.65–0.84) and “never” (OR = 0.72, C.I. 0.6–0.88). Regarding the consumption of sweets, compared with those who declared consumption “more than once a day”, the likelihood of association with a satisfactory perceived health status showed the highest odds of associations, with a statistical significance among all classes of consumption (“Sometimes a week” OR = 1.32 C.I. 1.07–1.62; Less than once a week OR = 1.26 C.I. 1.03–1.55). Again, considering the consumption of cured meat, the likelihood of association with a satisfactory perceived health status showed the highest odds of associations, with a statistical significance all among decreasing classes of consumption (“Once a day” OR = 1.45 C.I. 1.12–1.87, “Sometimes a week” OR = 1.31 C.I. 1.03–1.66; “Less than once a week” OR = 1.30 C.I. 1.02–1.66).

(iv)Logistic regression for obesity

We then performed a multilevel regression analysis with obesity as the main outcome (Table 4) and possible variables that were independently associated with it: consumption of TW showed a statistically significant low likelihood association (OR = 0.93, C.I. 0.87–0.99) with being obese. The obese status also showed an increasing trend of likelihood with aging, and it reached statistical significance among all different age classes (20–44 years, OR = 2.44, C.I. 1.68–3.55; 45–59, OR = 4.36, C.I. 3.01–6.33; 60–74, OR = 5.70, C.I. 3.92–8.29; ≥75 OR = 3.93, C.I. 2.68–5.78). The same trend was found regarding lower levels of education, which was statistically significant for high school diplomas (OR = 1.53, C.I. 1.36–1.71), middle school diplomas (OR = 1.78, C.I. 1.57–2.00), and elementary school diploma/none diploma (OR = 2.18, C.I. 1.89–2.50). An increasing trend of association for being obese in southern parts of Italy, compared to northern ones, was found but only the south geographical area reached a statically significant association (OR = 1.33, C.I. 1.21–1.47). A satisfactory perceived health status was negatively associated with the likelihood of being obese (OR = 0.63, C.I. 0.59–0.68) as well as alcohol consumption (OR = 0.83, C.I. 0.78–0.91). Considering the consumption of sweets, a low likelihood of an obese BMI was associated with lowest classes of consumption (“Once a day” OR = 0.71 C.I. 0.55–0.90, “Sometimes a week” OR = 0.77 C.I. 0.61–0.97). Regarding the consumption of carbonated drinks (excluding water), compared to those who declared themselves as consuming more than 1 L per day, all other subjects showed a negative association trend with being obese with the “never” consumption class, which reached statistical significance (OR = 0.67 C.I. 0.48–0.93). 

## 4. Discussion

Safe drinking water represents a worldwide major concern because of several factors, such as pollution and climate change, that are heavily impacting human health all around the globe, in particular among residents of developing countries. Impacts on surface water and groundwater resources and water-related illnesses are increasing, especially under changing climate scenarios such as diversity in rainfall patterns, increasing temperature, flash floods, severe droughts, heat waves, and heavy precipitation [29]. Recently, a link between health and environmental impacts and drinking water choices has been shown, since it has been estimated that the environmental impact of BW is 1400–3500 percent higher than TW [30]. On the other hand, BW consumption has sharply increased in recent years worldwide [31], and, interestingly, the recent increase in BW use globally has been driven by an increase in demand in low- and middle-income countries (LMICs), despite parallel increases in access to piped water in some countries [32]. To better understand these trends, it is extremely important to study possible variables associated with increased trap water consumption that can allow policy decision-makers to take adequate actions. Our study fits in this context, since we investigated several demographic and socioeconomic variables that can be associated with increased consumption of TW. Unlike other studies [13,15], our regression model did not observe a statistically significant association of TW consumption with gender; on the other hand, regarding age, the younger the age classes were less likely to be TW consumers, in particular people aged 20–44 years old. This result agrees with other studies [14]. Interestingly, lower levels of schooling compared to the graduate/post-graduate level showed a reduced frequency of TW consumption and this is partially in disagreement with other findings [33]. It is possible that in our country, people with higher levels of schooling have an increased awareness about the safety and importance of consuming TW also for environmental issues, as testified by our result in terms of the presence of concern about waste production, since we found a statistically significant association of increased likelihood of TW consumption for this variable. So, generally speaking, an increased level of knowledge may be associated with increased awareness and environmentally friendly behaviors. It must be highlighted that concern for climate change showed a similar trend, but it did not reach statistical significance in terms of association with TW consumption, indicating a complex scenario in terms of knowledge of environmental issues by the general population. An important consideration regards the Geographical area of residency: in the regression model, we observed a statistically significant difference among different geographical areas and we found a decreasing trend of TW consumption from North to South, particularly evident on the islands where there is the lowest likelihood of TW consumption. This result may be explained by the lower presence of natural water sources between the northern and the southern parts of Italy, especially for islands like Sicily and Sardinia where water is particularly scarce [34]. Looking at heating habits, TW consumption was shown to be associated with more healthy habits like a lower consumption of sweets, chicken and beef meat, possibly reflecting subjects’ awareness of a balanced healthy diet associated with consumption of non-bottled water.

Looking at health outcomes, we focused on the perceived health status declared by subjects and we studied the possible association with a positive attitude and consumption of TW together with other variables included in the logistic model. Even if TW consumption showed an increased likelihood of association with a satisfactory perceived health status, the association did not reach statistical significance. On the other hand, other variables showed a coherent association, in agreement also with studies conducted by others, since the likelihood of perceiving a satisfactory health status decreases with age [35], with underweight or overweight/obese BMI classes [36], with lower availability of economic resources [37] and lower levels of education [38,39]. Moreover, we found a worsening perception of own health status from North to south, as found by others [40,41]. Also, the consumption of vegetables showed interesting trends, with a low likelihood of a satisfactory perceived health status associated with a lower consumption of vegetables, which reached statical significance for the lowest classes (identified as “less than once a week” and “never”). These results are in accordance with other studies [42,43]. Consumption of sweets and cured meat both showed a higher association with a satisfactory perceived health status, associated with lower consumption classes [44,45]. Taken together, these considerations may allow us to speculate that the model proposed describes quite accurately the reality of variables independently associated with a satisfactory perceived health level, including the consumption of TW, which showed a positive likelihood of association even if not statically significant.

Looking at BMI, we found that the consumption of TW showed a possible protective effect towards being obese. This could be explained by the fact that individuals who preferred to consume TW were more likely to have both a healthier lifestyle and a lower consumption of refined, carbonated, and sugary beverages [22,46], which are responsible for weight gain [47], cardiovascular issues [48], diabetes [49] and overall poorer glycemic control [50]. Also, this model showed coherent associations with other independent variables, that were confirmed by other studies, since we found an increased likelihood of being obese with aging [51,52], with a lower educational level [52,53], for subjects with a geographical area of residency in the southern parts of Italy [54], with sweet consumption [45] and also with the consumption of carbonated drinks [47].

To our knowledge, this is the first study in Italy to take into consideration the consumption of TW and possible associated sociodemographic variables and health outcomes, such as perceived health status and BMI. Moreover, our data covered a nationwide sample of thousands of subjects, and interviews and data collection were conducted under rigorous methodological methods, since they were carried out by ISTAT-trained personnel. 

Nevertheless, some limitations should be acknowledged. Given the nature of this study, as a prevalence study rather than a longitudinal one, certain details regarding individuals’ dietary habits might have lacked precision. This is testified, for example, by the absence of the exact amount of TW consumption, which could have been more informative about water attitudes. Also, we did not have specific information on possible specific pathological conditions that could have been associated with the consumption of TW, but only aggregated variables, like the perceived health status, which we used in our models. Moreover, all data must be considered declarative, generating a possible declarative or recall bias in our sample. Lastly, it is important to note that the official language of ISTAT, the government institution conducting the survey, is Italian. The survey administration was not declared in other languages, which may have resulted in a selection bias. This bias could have excluded individuals, even residents in Italy, who did not fully understand the Italian language.

## 5. Conclusions

Taken together, our data show very informative results that can be very useful in terms of public health. For example, increasing education about the importance of TW consumption may improve this aspect among those lower-educated people who do not drink TW, maybe due to misleading beliefs about its safety, and at the same time increase a probably low self-awareness about the importance of plastic-free water consumption towards environmental issues that are related to BW production and distribution. Moreover, the increasing number of people accessing safe TW can be considered as a “One Health” approach, since through the administration of a fundamental health element and right such as free drinkable water, a nation can obtain important results on other several aspects like environmental issues and economic and social equality. Interestingly, some diet patterns may influence or be associated with TW consumption, as highlighted by our results. TW water can also influence health outcomes, both perceived and objective ones: we decided to focus on two general aspects of health outcomes, i.e., perceived health status and BMI, based on the availability of our data. Regarding BMI, according to already-existing studies, we investigated the association of TW consumption and an obese BMI, since numerous studies in the literature have explored enhancements of water consumption in relationships with cardiovascular diseases such as diabetes, hypertension, and obesity. Even if underweight could have been investigated, the representative sample with this condition in our population was too low and this BMI class is less frequent and problematic in developed countries like Italy. A possible further investigation could include more specific health outcomes such as specific chronic diseases and the consumption of TW, maybe with dedicated cross-sectional and longitudinal investigations. This approach may help health policymakers to better design health promotion campaigns and prevention strategies aimed at reducing health inequalities, which may be obtained also through the promotion of TW consumption.

## Figures and Tables

**Table 1 nutrients-16-00944-t001:** Distribution of the proportion of people drinking TW according to selected personal characteristics.

Variables		N	% of People Drinking TW	*p*-Value
Sex	Males	9524	43.28	0.58
	Females	10,150	43.02	
Age class (years)	<20	3142	15.97	0.00
	20–44	4577	23.26	
	45–59	4942	25.12	
	60–74	4298	21.85	
	≥75	2715	13.80	
Educational level	Graduate/Post-Graduate degree	3111	48.56	0.00
	High school diploma	6640	43.55	
	Middle school diploma	4874	41.19	
	Primary school diploma/none	4013	41.52	
Economical resources in the last 12 months	Optimal	356	50.00	0.00
	Adequate	13,545	43.76	
	Scarce	5103	41.24.	
	Inadequate	670	42.92	
Concern about waste production	No	11,711	42.45	0.00
	Yes	7963	44.21	
Concern about climate change	No	10,088	41.8	0.00
	Yes	9586	44.66	
Geographical Area of residency	North-West	4712	46.21	0.00
	North-East	5966	64.98	
	Center	3649	41.63	
	South	4482	35.8	
	The Islands	857	17.45	
BMI	Normal	8571	43.94	0.00
	Underweight	509	46.27	
	Overweight	5869	42.90	
	Obese	1974	40.66	
Carbonated drink consumption (excluding water)	More than 1 L/day	170	40.57	0.00
	From 1/2 L to 1 L/day	482	43.19	
	Less than 1/2 L day	1241	40.68	
	Occasionally	6103	41.89	
	Less than occasionally	1932	45.87	
	No/never	7897	44.96	
Alcohol consumption	No	5618	40.89	0.00
	Yes	12,519	44.66	
Consumption of vegetables	More than once a day	3584	46.83	0.00
	Once a day	6645	44.43	
	Sometimes a week	6968	41.80	
	Less than once a week	1349	39.20	
	Never	643	39.30	
Consumption of snacks	More than once a day	186	41.24	0.00
	Once a day	731	42.65	
	Sometimes a week	4276	40.71	
	Less than once a week	7265	45.10	
	Never	6616	43.50	
Self-perceived health status	Not satisfactory	3088	41.05	0.00
	Satisfactory	14,323	44.03	
Consumption of dairy products	More than once a day	910	48.74	0.00
	Once a day	3585	46.82	
	Sometimes a week	11,065	43.00	
	Less than once a week	2737	39.90	
	Never	874	40.26	
Consumption of sweets	More than once a day	445	40.20	0.25
	Once a day	2062	43.08	
	Sometimes a week	7119	43.34	
	Less than once a week	7372	43.68	
	Never	2135	43.37	
Consumption of cured meat	More than once a day	363	43.21	0.01
	Once a day	1450	43.23	
	Sometimes a week	9668	42.59	
	Less than once a week	5363	44.58	
	Never	1972	43.27	
Consumption of chicken meat	More than once a day	325	37.57	0.00
	Once a day	1650	40.77	
	Sometimes a week	13,074	42.38	
	Less than once a week	3393	48.5	
	Never	734	46.66	
Consumption of sheep meat	More than once a day	148	44.44	0.00
	Once a day	396	41.08	
	Sometimes a week	3796	38.6	
	Less than once a week	5600	43.34	
	Never	9126	45.78	
Consumption of beef meat	More than once a day	183	39.7	0.00
	Once a day	798	42.33	
	Sometimes a week	10,206	41.14	
	Less than once a week	6470	47.15	
	Never	1462	44.74	
Consumption of pork meat	More than once a day	166	43.34	0.00
	Once a day	587	41.16	
	Sometimes a week	7535	41.69	
	Less than once a week	7690	44.99	
	Never	3118	43.97	

**Table 2 nutrients-16-00944-t002:** Factors associated with TW consumption at logistic regression analysis. Hosmer–Lemeshow test 0.43.

Variables		OR	CI 95%	*p*-Value
Age	<20	1		
	20–44	0.75	0.65–0.88	0.00
	45–59	0.85	0.73–0.99	0.04
	60–74	0.86	0.73–1.00	0.06
	≥75	0.95	0.80–1.12	0.53
Educational level	Graduate/Post-graduate degree	1		
	High School diploma	0.8	0.75–0.85	0.00
	Middle school diploma	0.71	0.66–0.77	0.00
	Primary school diploma/none	0.69	0.63–0.76	0.00
Concern about waste production	No	1		
	Yes	1.09	1.04–1.14	0.00
Geographical Area of residency	North-West	1		
	North-East	2.13	1.99–2.27	0.00
	Center	0.85	0.80–0.91	0.00
	South	0.67	0.63–0.72	0.00
	The Islands	0.25	0.22–0.27	0.00
Alcohol consumption	No	1		
	Yes	1.11	1.05–1.17	0.00
Dairy Products	More than once a day	1		
	Once a day	0.87	0.77–0.99	0.03
	Sometimes a week	0.77	0.68–0.87	0.00
	Less than once a week	0.66	0.58–0.75	0.00
	Never	0.71	0.61–0.83	0.00
Sweet	More than once a day	1		
	Once a day	1.09	0.91–1.30	0.36
	Sometimes a week	1.18	1.00–1.40	0.05
	Less than once a week	1.22	1.02–1.44	0.02
	Never	1.27	1.06–1.52	0.00
Chicken Meat	More than once a day	1		
	Once a day	1.07	0.86–1.34	0.54
	Sometimes a week	1.20	0.97–1.49	0.09
	Less than once a week	1.45	1.16–1.81	0.00
	Never	1.33	1.04–1.70	0.03
Sheep Meat	More than once a day	1		
	Once a day	0.62	0.40–0.96	0.03
	Sometimes a week	0.60	0.39–0.90	0.01
	Less than once a week	0.59	0.39–0.89	0.01
	Never	0.54	0.36–0.82	0.00
Beef Meat	More than once a day	1		
	Once a day	1.67	1.12–2.49	0.01
	Sometimes a week	1.41	0.96–2.08	0.08
	Less than once a week	1.67	1.13–2.45	0.01
	Never	1.58	1.06–2.33	0.02

**Table 3 nutrients-16-00944-t003:** Factors associated with a satisfactory self-perceived health status at logistic regression analysis. Hosmer–Lemeshow test 0.08.

Variables		OR	95% CI	*p*-Value
Sex	Males	1		
	Females	0.83	0.78–0.88	0.00
Age	<20	1		
	20–44	0.65	0.49–0.88	0.00
	45–59	0.34	0.25–0.45	0.00
	60–74	0.21	0.16–0.28	0.00
	≥75	0.12	0.08–0.15	0.00
Educational level	Graduate/Post-graduate degree	1		
	High School diploma	0.96	0.87–1.05	0.38
	Middle school diploma	0.90	0.81–0.99	0.05
	Elementary school diploma/none	0.66	0.59–0.74	0.00
Economical resources in the last 12 months	Optimal	1		
	Adequate	1.01	0.78–1.31	0.94
	Scarce	0.51	0.39–0.66	0.00
	Inadequate	0.33	0.25–0.45	0.00
Concern about waste production	No	1		
	Yes	1.07	1.01–1.13	0.03
Geographical Area of residency	North-West	1		
	North-East	1.03	0.94–1.13	0.56
	Center	0.88	0.8–0.96	0.00
	South	0.8	0.73–0.87	0.00
	Islands	0.73	0.66–0.81	0.00
BMI	Normal	1		
	Underweight	1.47	1.23–1.75	0.00
	Overweight	1.32	1.10–1.58	0.00
	Obese	0.88	0.73–1.06	0.18
Alcohol consumption	No	1		
	Yes	1.32	1.24–1.41	0.00
Consumption of vegetables	More than once a day	1		
	Once a day	0.95	0.87–1.03	0.19
	Sometimes a week	0.88	0.81–0.96	0.00
	Less than once a week	0.74	0.65–0.84	0.00
	Never	0.72	0.6–0.88	0.001
Sweet	More than once a day	1		
	Once a day	1.11	0.90–1.38	0.33
	Sometimes a week	1.32	1.07–1.62	0.00
	Less than once a week	1.26	1.03–1.55	0.03
	Never	1.04	0.84–1.28	0.75
Cured Meat	More than once a day	1		
	Once a day	1.45	1.12–1.87	0.00
	Sometimes a week	1.31	1.03–1.66	0.03
	Less than once a week	1.30	1.02–1.66	0.03
	Never	1.23	0.95–1.59	0.11

**Table 4 nutrients-16-00944-t004:** Factors associated with obese BMI at logistic regression analysis. Hosmer–Lemeshow test 0.71.

Variables		OR	CI 95%	*p*-Value
Tap Water consumption	No	1		
	Yes	0.93	0.87–0.99	0.05
Sex	Males	1		
	Females	0.79	0.73–0.84	0.00
Age	<20	1		
	20–44	2.44	1.68–3.55	0.00
	45–59	4.36	3.01–6.33	0.00
	60–74	5.70	3.92–8.29	0.00
	≥75	3.93	2.68–5.78	0.00
Educational level	Graduate/Post-graduate			
	High School diploma	1.53	1.36–1.71	0.00
	Middle school diploma	1.78	1.57–2.00	0.00
	Primary school diploma/none	2.18	1.89–2.50	0.00
Geographical Area of residence	North-West	1		
	North-East	1.10	0.99–1.23	0.06
	Center	0.97	0.88–1.08	0.63
	South	1.33	1.21–1.47	0.00
	The Islands	1.12	0.99–1.27	0.08
Alcohol consumption	No	1		
	Yes	0.84	0.78–0.91	0.00
Self-perceived health status	Not Satisfactory	1		
	Satisfactory	0.63	0.59–0.68	0.00
Carbonated drinks consumption (excluding water)	More than 1 L/day	1		
	From 1/2 L to 1 L/day	0.73	0.5–1.06	0.1
	Less than 1/2 L day	0.90	0.64–1.27	0.55
	Occasionally	0.86	0.62–1.19	0.37
	Less than occasionally	0.73	0.52–1.03	0.07
	No/never	0.67	0.48–0.93	0.02
Sweet	More than once a day	1		
	Once a day	0.71	0.55–0.90	0.00
	Sometimes a week	0.77	0.61–0.97	0.03
	Less than once a week	0.84	0.67–1.06	0.14
	Never	0.85	0.67–1.09	0.20

## Data Availability

The data undergo anonymization and aggregation by ISTAT and are subsequently made publicly accessible for research purposes at the following link: https://www.istat.it/it/archivio/129956 (access on 12 December 2023).

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
