# Peer review of "Consumption of Tap Water and Sociodemographic-Associated Characteristics: A Nationwide Cross-Sectional Study"

_nutrients, 2024, doi:10.3390/nu16070944_

Round 1

Reviewer 1 Report

Comments and Suggestions for Authors

The authors aimed to examine tap water consumption, assessing associated sociodemographic factors and related health outcomes such as obesity and self-perceived health status at a nationwide level in Italy. I found this manuscript to be an interesting source of information. I have some comments to add in case they are useful:

General Comments

=================

This paper, in broad terms, has found some sociodemographic and health-related factors associated with tap water consumption. From my point of view, this paper needs improvement in certain aspects.

Major Comments

INTRODUCTION SECTION

The introduction is well-written and pleasant to read. I would like to alert the authors that I have detected poor editing in the Introduction section. For instance, a period is missing (line 43), and there are some extra spaces between words. On the other hand, in order to further enhance this section, I would add some sentences about the study variables (sociodemographic and health-related) from line 53 onward, as the authors have only mentioned gender differences and self-perceived health status

METHODS SECTION

I think the Materials and Methods section could be presented more effectively by dividing it into subsections. For example: (i) Design, Data Source, and Participants, (ii) Variables, (iii) Ethical Considerations, and (iv) Statistical Analysis. I have been surprised by the following:

I have not been able to understand the inclusion of the variables, such as consumption of vegetables or snacks. Why have other variables such as meat consumption, sweets, dairy products, etc., not been included, for example?

RESULTS SECTION

Similarly, I would divide the Results section into subsections.

I have read the article thoroughly, and I still don't quite understand why a separate logistic regression was performed with the "satisfactory self-perceived health status" variable and not with other variables (for example, underweight). You need to focus on the study objectives.

In Tables 3 and 4, what test was used to assess the goodness of fit of the logistic regression model?

CONCLUSIONS SECTION

I would revise the conclusions section in light of the reviewers' suggestions. This section can be improved.

Minor Comments

1.     In Introduction section, please, include a bibliographic citation in the text and its corresponding reference in “References” section at the end of the sentence: “Anyway, in the next future, the availability of safe water could be a worldwide issue”.

2.     In Introduction section, I have detected a small error (repeated bibliographical citations) in line 40, specifically at the end of the sentence: "Many studies have shown how the production and consumption of bottled water (BW) are contributors to these problems [3,4] s[3,4]".

3.     In Introduction section, please, include a bibliographic citation in the text and its corresponding reference in “References” section at the end of the sentence: “The European Region has diverse TW consumption patterns between and within countries, with the highest share of TW intake observed in northern Europe”.

4.     In the Materials and Methods section, please include a bibliographic citation in the text and its corresponding reference in the “References” section regarding the questionnaire “Aspects of daily life”, conducted by the Italian National Statistics Institute (ISTAT). I would ask the authors to, if possible, reference the methodology employed by ISTAT in conducting this questionnaire.

5.     In the Materials and Methods section, I would use the term “gender” rather than “sex”.

6.     In the Materials and Methods section, particularly, in the “Body Mass Index” variable, I would precisely indicate the kg/m2 for each subtype: underweight, normal weight, overweight, and obese.

7.     In the Materials and Methods section, I would create a separate subsection for Statistical analysis.

8.     In the Materials and Methods section, I would ask the authors to include a summary of what they have stated in the “Institutional Review Board Statement” section in a subsection titled “Ethical Consideration”.

9.     In Results section, particularly, in Table 1, you use “p”, and in Tables 2 and 3, you use “p-value”.

Reviewer 2 Report

Comments and Suggestions for Authors

Dear Authors,

I thank the Editor for entrusting me to review this manuscript. Drinking the right quantity and quality is extremely important for the proper functioning of the human body. Providing it to the world population is a major challenge globally especially in the face of its scarcity and pollution. In addition, the production of bottled drinking water causes further problems related to its recycling and the production of excessive waste. The solution to this problem would be to have such safe water available in waterworks to replace bottled water. Such a situation would benefit the environment. I congratulate the authors for undertaking a study on tap water (TW) consumption.

Below are my suggestions / comments:

The analysis of TW consumption was conducted by ISTAT on 20,000 families and 45597 individuals. A study conducted on such a large representative sample provides ample opportunity for inference and generalization of results.

Which level of the dependent variable in the model was predicted?

In Table 2, I propose to include only statistically significant variables in the logistic regression model (statistically insignificant variables have no effect on the dependent variable).

In the model for satisfactory Self-perceived health status, indicate which state the dependent variable was predicted.

In all models, independent variables not statistically significant in the model should not be presented and analyzed.

Round 2

Reviewer 1 Report

Comments and Suggestions for Authors

The authors have followed my suggestions, and I recommend this paper for publication in Nutrients.